## Research Article

young migrants; suicide; social media

**Corresponding author:**
Aditya Basu;
Email: basu@student.unimelb.edu.au

# A qualitative study of young migrants' encounters with suicide and self-harm content, connecting and seeking help on social media

Aditya Basu[1,2] ⓘ, Greg Armstrong[3], Michelle Lamblin[4,5], Jo Robinson[5,6] and Samuel McKay[5,6]

[1]Centre for Youth Mental Health, The University of Melbourne Faculty of Medicine Dentistry and Health Sciences, Australia; [2]Suicide Prevention Research, Orygen The National Centre of Excellence in Youth Mental Health, Australia; [3]The University of Melbourne Nossal Institute for Global Health, Australia; [4]Orygen Limited, Australia; [5]The University of Melbourne Centre for Youth Mental Health, Australia and [6]Orygen The National Centre of Excellence in Youth Mental Health, Australia

## Abstract

Young migrants encounter heightened challenges as the intersection of their youth and migrant identities magnifies the influence of risk factors for suicide. Social media offers a platform for young migrants to express emotions, seek support and connect with others, often anonymously. However, how they communicate about self-harm and suicide online remains underexplored. This qualitative study involved 17 online interviews with young migrants aged 15–25 years. Transcribed data were coded and thematically explored using Braun and Clarke's reflexive thematic analysis methodology. Four key themes were identified: (1) Exposed and isolated: The emotional toll of viewing self-harm and suicide-related content on young migrants; (2) Connected but at-risk: The dual role of social media in migrant belonging; (3) Digital belonging across borders: Benefits and strains of staying connected; and (4) Helpful and harmful: The dual nature of support on social media. Social media has a complex impact on young migrants, offering both protective and harmful effects. While exposure to distressing or discriminatory content may exacerbate feelings of isolation and disconnection, social media can also promote belonging, cultural understanding and resilience. It also provides accessible support, though poor-quality advice and stigma may deter help-seeking. These insights can inform culturally responsive mental health interventions.

## Impact statement

Young migrants often turn to social media to find connections, express emotions and cope with the challenges of adjusting to a new country. This study offers new insights into how they encounter and respond to self-harm and suicide-related content online. By listening directly to young migrants, the research highlights the emotional strain caused by unexpected exposure to distressing material, including self-harm and suicide-related content and discriminatory posts. These experiences can intensify feelings of loneliness and exclusion, especially for those who are already navigating complex migration journeys. At the same time, the study shows that social media can play an important positive role. Many young migrants use these platforms to build friendships, learn about cultural expectations in their host country and maintain meaningful ties with people and traditions from their country of origin. Online spaces can provide a sense of safety and anonymity, allowing young migrants to discuss their struggles or look for support without fear of judgement. However, the study also reveals the risks of misinformation and poor-quality advice, which may discourage help-seeking or contribute to confusion. The findings point to a need for mental health resources and online safety strategies that recognise the specific experiences of young migrants. Policymakers, educators and mental healthcare providers can use this evidence to design more culturally aware interventions and to develop safer online environments. Social media companies can also draw on these insights to improve content moderation and to create tools that better protect young people from harmful material. Overall, this research shows that young migrants' online experiences are shaped by both vulnerability and resilience. Supporting them effectively requires an approach that acknowledges the realities of their digital lives and promotes inclusive, culturally responsive forms of mental health support.





## Introduction

In Australia, suicide is the leading cause of death among people aged between 15 and 25 years, with deaths by suicide representing approximately one-third of all deaths in this age group

(Australian Institute of Health and Welfare, 2025). Self-harm occurs more frequently and is a predictor of future suicidal thoughts and behaviours (Chan et al., 2016). Rates of hospitalisation due to self-harm have increased in Australia over time (Australian Institute of Health and Welfare, 2025). A migrant is defined by the International Organization for Migration (IOM) as any person who has moved across an international border away from their habitual place of residence, regardless of the person's legal status, whether the movement is voluntary or involuntary, what the causes for the movement are or what the length of the stay is (International Organization for Migration (IOM), 2024). Migrants make up a substantial portion of the Australian population and face several unique risk and protective factors for suicide. For instance, migrants may experience feelings of loneliness after moving from one country to another, limited proficiency in the host country's primary language or reduced participation in community activities and events (Ten Kate et al., 2020; Delaruelle, 2023). Conversely, migrants' cultural beliefs may protect against self-harm and suicide, such as transparently sharing stressful life experiences with community members, ties with cultural values and practices and the belief that suicide is amoral (Chu et al., 2010). Young migrants face intensified challenges as their youth and migrant identities overlap, amplifying the interplay of risk and protective factors (Moran et al., 2024). The term "young migrants" refers to migrants aged between 15 and 25 years and encompasses diverse groups with distinct migration trajectories, such as international students, refugees and asylum seekers, those who had migrated with family members at a young age and those who had migrated for employment opportunities (Vertovec, 2007). Some young migrants may need to negotiate their societal identity in their host country, which may contradict their identity in their country of origin, resulting in feelings of disconnection and isolation (Mude and Mwanri, 2020; Binder et al., 2025).

The Interpersonal Theory of Suicide posits that feelings of disconnection and isolation can result in thwarted belongingness – a risk factor for suicide (Van Orden et al., 2010). This is also reiterated by the Cultural Theory of Suicide's concept of social discord, which explains that alienation from family and community members is a risk factor for suicide (Chu et al., 2010). Some migrants may experience discrimination and changes in social roles and socio-economic status (Forte et al., 2018; Maheen et al., 2024). This may result in minority stress (stresses cultural minorities experience because of their social identity or position), a risk factor for suicide as defined by the Cultural Theory of Suicide. Limited research on this topic found that young migrants experience elevated rates of self-harm and suicide attempt, with no major differences in suicidal ideation and suicide death compared to non-migrant young people (Basu et al., 2022).

Reasons for self-harm and suicide are complex; however, social media use is often cited as an influencing factor (Memon et al., 2018). Exposure to certain types of content, such as depictions of suicide or self-harm, may contribute to psychological distress and increase an individual's likelihood of engaging in similar behaviour (Susi et al., 2023). Young people aged between 15 and 25 years are more likely to be exposed to self-harm and suicide-related content on social media compared to adults aged 26 years and over (Robinson et al., 2024). This is also true for young migrants aged between 15 and 25 years who are more likely to be exposed to such content compared to older migrants. Young people of both migrant and non-migrant backgrounds use social media to access support for self-harm and suicide online (Robinson et al., 2024). This is often because it offers free support that is available throughout the

day, overcoming financial and geographic barriers to access, or in some cases, anonymous support, which can reduce potential stigma and facilitate disclosure of distress (Naslund et al., 2016). Social media helps migrants find belonging with others sharing their cultural or migration background, protecting against acculturative stressors like anxiety, loneliness and isolation – key suicide risk factors. A sense of belonging can encourage young people to seek help for self-harm and suicidal thoughts, as it fosters trust and reduces feelings of isolation that often act as barriers to help-seeking (Chu et al., 2018). This suggests that social media's impact on self-harm and suicide is complex – neither entirely harmful nor helpful – and can vary depending on the individual and their emotional state at the time (Thorn et al., 2023).

It is important to acknowledge that social media platforms operate through commercial logics that commodify attention and emotional engagement, shaping what content is circulated and prioritised for users (Carrasco et al., 2025). For example, algorithms tailor content to user behaviour (Eg et al., 2023) – meaning someone who has sought help for self-harm or suicide may continue to be shown related content, even without actively seeking it. Additionally, platform-specific content flows, such as trending topics, using hashtags and peer sharing, can facilitate rapid dissemination and normalisation of harmful content, complicating efforts to moderate or interrupt exposure pathways (Arendt et al., 2019). This structural lens helps explain how algorithmic amplification, monetisation incentives and platform economies may disproportionately expose young migrants to harmful content, particularly when their vulnerabilities intersect with platform engagement metrics.

A national study examined how often young Australians (aged 15–25 years) and adults (26 years and over) are exposed to self-harm and suicide-related content on social media, whether they create such content themselves, and how they use social media to seek support for self-harm and suicide (Robinson et al., 2024). The results were analysed to examine responses for migrants, which showed that young migrants are exposed to self-harm and suicide-related content and use social media to seek help for self-harm and suicide. This study aims to build on these findings to explore young migrants' perceptions of (i) being exposed to self-harm and suicide-related content on social media; (ii) using social media to form and maintain connections; and (iii) seeking support for self-harm and suicide online. Examining these factors, from the perspective of the young migrants, is necessary to develop policies and technologies that can optimise the benefits of social media and minimise the risks related to self-harm and suicide. This may provide opportunities to reduce self-harm and suicide among young migrants while helping to build and sustain safer online spaces.

## Methods

### Research questions

The study aims to explore the following research questions:

1. What are the perceived impacts of encountering self-harm and suicide-related content on social media among young migrants?
2. How do young migrants use social media to form connections with people in Australia and maintain connections with people and cultural practices in their country of origin?
3. What are young migrants' experiences of seeking support for self-harm and suicide on social media?

## Study design

This qualitative study utilised semi-structured interviews with young migrants. Interview questions (Supplementary Appendix A) were open-ended and developed specifically for this study based on the findings of previous literature and the results of the survey on which this study was built. The study has been reported in line with the Consolidated Criteria for Reporting Qualitative Research (COREQ) (Tong et al., 2007) (Supplementary Appendix B).

## Researcher preparation and reflexivity

The lead author and interviewer (AB) undertook suicide intervention skills training and qualitative research training before commencing the interviews. Several co-authors contributing to the study brought a distinctive perspective shaped by their experiences as a migrant, AB, having moved multiple times as a young person, and SM, having experience in multiple countries as an international student. This insider status, as outlined by Dwyer and Buckle (2009), brought a unique perspective to the analysis, shaped by personal experience of migration among young people, which informed how the researchers viewed, interpreted and reflected on the data. This lens added a layer of depth and nuance, an additional depth that someone with first-hand experience of migration as a young person can provide when exploring experiences of other young migrants. The authors acknowledged that using an insider lens allowed deeper empathy and understanding but risked bias from their own backgrounds. To mitigate this, they maintained critical self-awareness, involved multiple team members in analysis and integrated findings with existing literature. For example, during coding, AB's recognition of culturally specific metaphors used by participants prompted team discussions that led to the refinement of thematic categories. Additionally, reflexive dialogues between AB and SM helped surface implicit assumptions (e.g., assuming that migration is a traumatic rupture and, therefore, a primary driver of suicidal distress, certain cultural backgrounds are characterised by stigma around mental health and suicide), ensuring that interpretations remained grounded in participants' narratives rather than researchers' personal experiences.

## Ethics and safety

The study received approval from the University of Melbourne Office of Research Ethics and Integrity (Reference number: Ref: 2023-27352-46587-3). All participants provided written consent, and parental consent was required for those under 18 years of age. Before each interview, participants provided an emergency contact and location in case of crisis. Afterward, the interviewer acknowledged potential distress and reminded them to reach out to healthcare professionals, sharing national helpline contacts.

## Sample and recruitment

Participants were recruited from a pool of individuals who had completed a national cross-sectional survey of the general Australian population's experiences and perceptions around creation of and exposure to self-harm and suicide content on social media (the methodology of recruitment for that survey has been reported in detail elsewhere). To be eligible to participate in this study, the participant had to be aged between 15 and 25 years (inclusive) and born outside of Australia. The lead author (AB) identified young migrant survey participants who agreed to follow-up, obtained informed consent (including parental consent for minors) and provided all participants with a Plain Language Statement outlining the research aims and AB's PhD involvement. Participants came from a variety of backgrounds, including people who had migrated with family for schooling, international students, people who had migrated for employment opportunities and those who had moved as a refugee or asylum seeker. This ensured that a variety of perspectives were obtained from the interviews, providing a breadth of responses.

## Data collection and analysis

Data were collected in August–September 2024. AB conducted 45–60 min interviews via Microsoft Teams, audio-recorded and transcribed using its built-in feature. AB reviewed and edited transcripts and took field notes. Participants received $45 AUD reimbursement.

Data were analysed using reflexive thematic analysis following the six steps outlined by Braun and Clarke: (1) familiarising with data, (2) generating an initial coding frame, (3) searching for themes, (4) reviewing themes, (5) defining and naming themes and (6) reporting. The lead author (AB) immersed in the transcripts, coded data inductively in NVivo 15, creating 21 codes grouped into themes. Co-author (SM) reviewed and refined codes, with regular consultation from GA and JR. Final themes were organised and illustrated with transcript quotes.

## Results

Seventeen young migrants were interviewed for this study. See Table 1 for demographic characteristics.

Four key themes were identified (see Table 2):

1. Exposed and isolated: The emotional toll of viewing self-harm and suicide-related content on young migrants
2. Connected but at-risk: The dual role of social media in migrant belonging

**Table 1.** Demographic characteristics of participants

| Demographics | Participants |
| --- | --- |
| *Gender* | |
| Man or male | 9 |
| Woman or female | 8 |
| *Age group* | |
| 15–17 years | 3 |
| 18–25 years | 14 |
| *Location of residence* | |
| Metropolitan/urban centre | 14 |
| Rural area | 3 |
| *Region of birth* | |
| Asia | 11 |
| Africa | 2 |
| Europe | 3 |
| Oceania (excluding Australia) | 1 |

## Theme 1: "Exposed and isolated: The emotional toll of viewing self-harm and suicide-related content on young migrants"

All participants reported having encountered self-harm and suicide-related content on social media, almost always accidentally rather than through deliberate searching. Most described these encounters as distressing, lowering their mood and prompting them to avoid similar content. As one explained, "*Normally I just try to skip past those posts as fast as I can… I don't want to expose myself to anything that's going to worsen my mental state*" [YM 17]. Another felt such content could be harmful to others, noting, "*A person shouldn't post this because it could maybe drive other people to do the same thing*" [YM 08]. This underscores how exposure to such content can negatively affect some young migrants' well-being and heighten concern for others' safety.

Repeated exposure sometimes led to desensitisation, with participants describing how frequent exposure normalised or dulled their emotional response to distressing material. One participant reflected: "*Unfortunately, being exposed to so much of it, you become*

**Table 2.** Overview of themes and their descriptors

| Theme | Description |
| --- | --- |
| *RQ1: What are the perceived impacts of encountering self-harm and suicide-related content on social media among young migrants?* | |
| Exposed and isolated: The emotional toll of viewing self-harm and suicide-related content on young migrants | Exposure to self-harm and suicide-related content can heighten emotional distress and fear for others' well-being, although it may also reduce feelings of isolation by validating personal experiences. Negative perceptions of those who share such content can further complicate emotional responses. |
| *RQ2: How do young migrants use social media to form connections with people in Australia and maintain connections with people and cultural practices in their country of origin?* | |
| Connected but at-risk: The dual role of social media in migrant belonging | Social media fosters cultural learning and connection for young migrants, supporting integration and belonging. However, encounters with racism and discrimination can undermine these benefits, reinforcing feelings of exclusion and vulnerability. |
| Digital belonging across borders: Benefits and strains of staying connected | Maintaining transnational connections through social media strengthens cultural identity and belonging. However, ongoing exposure to distressing news or crises from home can evoke feelings of isolation and emotional strain. |
| *RQ3: What are young migrants' experiences of seeking support for self-harm and suicide on social media?* | |
| Helpful and harmful: The dual nature of support on social media | Online platforms offer accessible support and a sense of solidarity around self-harm and suicide. At the same time, unmoderated advice and fear of stigma may deter help-seeking. |

*a little bit desensitised… It just seemed like a part of growing up or a part of going through puberty or adolescence*" [YM 05].

At the same time, some participants felt conflicted when exposure evoked both empathy and helplessness. Seeing self-harm framed as a form of disclosure or expression left them concerned but powerless to act: "*She posted a video saying it was OK to show your scars, then removed her hoodie to reveal arms covered in them… It was upsetting because I felt sad because I was unable to help her*" [YM 13].

Two participants actively searched for content, usually to feel less alone. For instance, once described using online forums to relate to others with similar struggles: "*I can read their stories. It kind of makes me feel like, hey, looks like I'm not the only person that's suffering*" [YM 08].

Others, however, questioned the motives of people posting self-harm or suicide-related content, sometimes interpreted as attention-seeking. One participant remarked, "*I believe some young people… pretty much just want attention… I think that's why they post stuff like that to social media*" [YM 02].

The only participant who stated that they had shared self-harm content themselves said they did so because they felt it made them appear more mature: "*At the time, I wanted to seem more grown up, so I reposted those images to fit in*" [YM 05]. Young migrants engaged with self-harm–related content in varied ways – some sought connection and reassurance, while others questioned the motives behind such posts or shared them to appear more mature. This highlights the complex social pressures shaping how young people interpret and interact with this material online.

## Theme 2: "Connected but at-risk: The dual role of social media in migrant belonging"

Participants felt that social media helped them to build connections and learn about Australian culture, aiding assimilation, but also exposed them to racist and discriminatory content that made some of them feel unwelcome.

Social media helped some participants connect with new people in Australia by passively learning about interests, news, humour and culture, enabling quicker, more informed conversations. "*Seeing cultural influences and humour on social media helps me stay connected and understand the Australian sense of humour, making it easier to chat with classmates and coworkers*" [YM 04].

Social media increased participants' awareness of friends' interests and events in Australia, enabling them to participate in shared experiences that strengthen bonds and reduce isolation. One participant stated: "*On Facebook, seeing friends show interest in events on my feed helps me discover things I didn't know about and consider attending*" [YM 15]. In this way, social media played a positive role by helping some young migrants feel more connected in Australia, giving them insights that made conversations easier and strengthened their sense of belonging.

In contrast, exposure to racist comments on social media caused some participants, including culturally Australian-identifying ones, to feel hurt, disconnected and question their belonging, showing the deep impact of discrimination on their connection to Australia. One participant reflected: "*Encountering racist content or comments in the news makes me feel personally offended and disconnected from Australia*" [YM 01]. Another remarked: "*When someone from your minority group does something viewed negatively, it can lead to generalisations, which makes me feel sad and less connected to Australia*" [YM 12]. Exposure to racist content on social media was experienced not as distant commentary but as a personal affront, fostering vigilance, self-doubt and a sense that their place in Australia

was conditional. The cumulative and algorithmically amplified nature of online hostility intensified feelings of alienation, shaping how safe and accepted participants felt both online and offline.

### Theme 3: "Digital belonging across borders: Benefits and strains of staying connected"

Some participants used social media to connect with people and cultural practices from their origin country, fostering connection and learning, but exposure to certain content also caused feelings of disconnection.

Participants reported that social media helps them to maintain connections with family and friends in their country of origin by providing content that prompts them to reach out and communicate with them. One remarked: "*When friends back home post about their day or visits, I like their posts and message them, which helps me feel included in their lives*" [YM 05]. Another example was: "*If I share a video with them, my family will just come together as one, and then we'll just talk all together… It gives us something to chat about*" [YM 01].

Social media helped participants reconnect with cultural practices and language, supporting their efforts to maintain cultural roots after migration. One participant explained: "*I use YouTube to learn traditional dances, helping me stay connected to my roots and makes me feel proud of my culture and history*" [YM 07].

Some participants felt disconnected and reluctant to return home after encountering negative content from their country of origin. One participant provided an example: "*In Samoa, rival villages sometimes clash violently, even with guns, and people often share videos of these conflicts on Facebook. It makes me feel worried and scared*" [YM 07]. Social media exposes young migrants to both favourable and unfavourable portrayals of their countries of origin, resulting in mixed effects on their sense of belonging.

### Theme 4: "Helpful and harmful: The dual nature of support on social media"

Some participants used social media for confidential support on self-harm and suicide, often connecting with those from similar cultural backgrounds. However, biased advice and cultural stigma sometimes hinder help-seeking and sharing distress online.

Some participants used social media to discuss self-harm or suicidal thoughts, valuing its accessibility, especially across time zones, as a reliable source of support when needed. "*If you have friends who are up late at night, you can always message them. Especially those in other time zones. It kind of makes you feel like you can always turn to someone to talk to*" [YM 12].

The text-based, asynchronous nature of communication allowed time for reflection and clearer communication, benefiting those who struggle with verbal or real-time conversations. This applied to participants seeking help from their peers as well as using mental healthcare support services. "*Using text lets me reflect and respond at my own pace, reducing anxiety by allowing thoughtful, one-question-at-a-time communication*" [YM 11]. The virtual nature of communication also helped some participants overcome their fear of being stigmatised, which they may experience when seeking support in-person. "*When I can message someone for help from my phone, it feels like I'm not being judged by them… But doing the same thing in-person would be scary for me*" [YM 03]. Online communication helped young migrants express themselves more clearly and seek support with less fear of judgement, offering a sense of safety that many struggled to find in real-time or in-person interactions.

Some participants preferred connecting with young migrants of similar backgrounds, feeling more comfortable and understood, enabling clearer discussions about self-harm and suicide without explaining cultural perspectives. One participant provided an example: "*In a mental health group of 10,000, I connected mainly with others of Asian descent because shared culture created understanding and safety, as we'd faced similar challenges and family attitudes toward mental health*" [YM 12]. This suggests that some young migrants felt safer and better understood when connecting with peers from similar cultural backgrounds, which enabled more open and nuanced conversations about sensitive topics without needing to justify their cultural perspectives.

However, the familiarity of cultural identity may also have unintended negative consequences. Some participants noted that advice on social media can unintentionally cause harm, as it may be subjective, culturally biased and not evidence-based. Well-meaning support, especially from older adults, might be unhelpful or hurtful to those seeking help if the person providing support already stigmatises mental health and suicide. "*My dad once shared a social media story about suicide linked to money problems, intending to help, but it felt inappropriate and confusing*" [YM 12]. This suggested that receiving advice from their parent about suicide had a negative impact on the participant because the person sharing the advice was biased.

## Discussion

### Key findings and implications

The study examined young migrants' views on social media and self-harm/suicide, revealing its dual role. Four themes were identified: (i) exposure to self-harm/suicide content has a negative impact on mood; (ii) social media fosters belonging yet can reinforce exclusion; (iii) it maintains cultural ties but may cause disconnection; and (iv) online support offers opportunities but risks misinformation and stigma.

Most participants described exposure to self-harm and suicide-related content on social media as distressing, typically accidental and harmful to emotional well-being. Evidence shows such exposure can increase the risk of suicidal thoughts, emotional disturbance and self-harming behaviour through a contagion effect (Arendt et al., 2019; Calvo et al., 2024), which is problematic given the high rates of accidental exposure among young people, including migrants (Robinson et al., 2024). Some participants also reported becoming desensitised from repeated encounters with this content, reflecting compassion or empathy fatigue that can undermine emotion regulation, empathy and hope (Stoewen, 2020; Roberts, 2021). These findings highlight the need for stronger safeguards that reduce exposure to self-harm- or suicide-related content on social media platforms through the development of policies in collaboration with users, healthcare providers and policymakers (Robinson et al., 2023).

Participants used social media to stay connected with family abroad and build new relationships in Australia. These connections may protect against 'thwarted belongingness' – a suicide risk factor marked by disconnection and isolation (Van Orden et al., 2010). A small minority of young migrants actively sought relatable content online, finding that connecting with others with similar experiences helps reduce loneliness and isolation (Coulson et al., 2017). By offering a platform to form and maintain connections, social media can protect young migrants from social discord (defined by the Cultural Theory of Suicide as alienation from family and community (Chu et al., 2010) and lowers suicide risk (Whitlock et al., 2014; Arango et al., 2024). Interacting with people of the same ethnic

background can provide migrants with emotional support, comfort, a sense of belonging and connectedness and buffer against feelings of loneliness (Rios Casas et al., 2020; Rüdel and Joly, 2024). Social media can help young migrants stay connected to their cultural practices, boosting self-esteem and mental health, which can reduce suicide risk (McGorry et al., 2024). It can also buffer against discrimination, racism and acculturative stress (i.e., stressors from adapting and functioning as a member of a non-dominant cultural group) (Marcelo and Yates, 2019; Gibson et al., 2021). This may help prevent cultural bereavement – grief and distress from losing cultural values, social ties and identity (Bhugra and Becker, 2005). Healthcare providers could consider social and cultural connectedness when caring for young migrants and the potential role of social media in facilitating a sense of connection and belonging.

Although social media helped participants to learn more about Australia, it can also expose them to potentially harmful content. Several participants reported that coming across racist or discriminatory comments resulted in them feeling disconnected from Australian society. Racial discrimination (both online and offline) has been associated with depressive symptoms and substance abuse among young people of ethnic minority backgrounds, both of which are risk factors for suicide (English et al., 2020; Keum and Cano, 2021). Young migrants encountering online racism report greater feelings of being a burden, which can lead to internalised racism and disconnection, and is a risk factor for suicide (Keum, 2023; Keum and Choi, 2024). The Cultural Theory of Suicide explains minority stress at two levels: distal (external events like racism on social media) and proximal (internalising negative stereotypes) (Chu et al., 2010), and racist comments affect young migrants at both levels. Some participants encountered content about their origin country that made them feel detached, increasing thwarted belongingness and suicide risk (Van Orden et al., 2010). Exposure to negative media or news from their country of origin can intensify migrants' feelings of cultural disconnection and alienation, which in turn is associated with increased psychological distress and poorer mental health outcomes (Sønderskov et al., 2021; Verelst et al., 2022). This emotional dissonance, heightened by a perceived inability to influence or change events in their homeland, may undermine a young migrant's sense of identity and belonging, further compounding risks of anxiety, depression and social withdrawal (Brance et al., 2024). Young migrants disconnected from both their origin and destination countries may experience marginalisation – rejecting both cultures – leading to poor mental health and higher suicide risk (Berry and Sam, 1997; Choy et al., 2021). Policymakers should implement frameworks to require social media platforms to remove racist content and undertake education campaigns to boost social media literacy among young migrants to help them critically assess and counter online negativity (Keum and Choi, 2024). Digital platforms can operationalise culturally responsive safety measures by integrating community-informed moderation practices that reflect the values, languages and norms of diverse user groups (Matamoros-Fernández, 2017). This includes adapting algorithms to recognise culturally specific expressions of distress and co-designing safety tools with marginalised communities to ensure relevance and trust (Gillespie, 2018).

Participants described using social media to access support confidentially and easily. Migrants face cultural (stigma, low mental health awareness), structural (financial, language), trust and confidentiality barriers when seeking help for self-harm and suicide (Krendl and Pescosolido, 2020; Schouler-Ocak et al., 2020; Sotardi, 2024). Participants emphasised confidentiality, saying they would only seek help through private messaging to ensure privacy and avoid stigma. Stigma and cultural beliefs play a major role in deterring migrants from seeking professional mental healthcare services compared to non-migrants (Place et al., 2021; Mohammadifirouzeh et al., 2023). This suggests that young migrants may experience unmet mental health needs and delayed intervention due to the fear of experiencing stigma. Social media can offer a confidential space to share experiences and seek support, reducing the fear of stigma. Anonymity on social media can reduce perceived stigma and fear of judgement, allowing young people to disclose mental health or suicidal struggles more freely and increasing their willingness to seek help (Pretorius et al., 2019). Compared to in-person discussions, young people are more willing to discuss topics such as mental health and suicide online, highlighting how being physically separated from the person may overcome barriers to seeking support (Wong et al., 2021). Some participants used social media to support peers, especially from their cultural background, which can help to reduce perceived burdensomeness and lower suicide risk (Rice et al., 2016). Connecting with others facing similar challenges on social media reduces loneliness, builds belonging and lowers suicide risk by breaking stigma (Chen and Wang, 2021; Thapliyal et al., 2024). Young migrants' use of social media for support underscores the need to optimise platforms for evidence-based, culturally sensitive content. Promoting social media as a safe space, supported by confidentiality policies, can raise awareness of suicide risk. Non-governmental organisations, particularly those supporting young people and migrants, should develop educational resources that are tailored for migrant communities, including by incorporating multilingual content (Orygen, 2026). Healthcare providers and health technology developers can collaborate to develop online platforms as a complement to in-person mental healthcare services.

Participants acknowledged that advice shared online can be influenced by cultural beliefs and may, at times, be unhelpful or harmful to the recipient, which can deter young migrants from seeking help (Samari et al., 2022). Furthermore, cultural beliefs deterred some participants from seeking support on social media because they feared being perceived as weak by their peers. Cultural beliefs may require a migrant to deal with challenges internally and thus stigmatise help-seeking (Byrow et al., 2020; Meribe et al., 2025). These findings show that social media offers both opportunities and challenges for suicide prevention. It should remain accessible for young migrants, with measures to ensure evidence-based advisory content, fact-checking, transparency and better digital literacy (Kington et al., 2021). Legislation controlling access to social media among young people (such as Australia banning access to those under 16 years) will reshape both the positive and negative impacts of social media – reducing exposure to harmful content and online risks while also potentially limiting opportunities for connection, digital literacy and access to mental healthcare services (Robinson et al., 2025).

### Strengths and limitations

Participants self-selected from a national online survey, likely reflecting those more engaged or knowledgeable about youth suicide/self-harm. This may have inadvertently excluded young migrants who avoid mental health topics due to stigma or cultural norms. Most were born in Asia and lived in urban Australia, limiting the applicability of findings to migrants from other regions or rural areas. Interviews were conducted in English and online, excluding those with limited English proficiency, digital literacy or internet access. Future research should purposefully recruit more

diverse migrant groups, including those from underrepresented regions and rural areas, and employ culturally sensitive, multilingual and community-based recruitment strategies (e.g., partnerships with cultural organisations and offline data collection methods) to engage young migrants who may be less connected to online mental health discourse, have limited English proficiency or digital access or experience stigma around discussing suicide and self-harm.

Despite limitations, the study offers valuable insights into young migrants' views on self-harm and suicide content on social media, a largely underexplored area. The qualitative, reflexive thematic approach enabled in-depth exploration and strengthened the transparency of results. Most co-authors brought extensive experience in suicide prevention research involving young migrants. Together, these strengths enhance the credibility and cultural sensitivity of the findings and provide a robust foundation for future research to further examine nuanced cultural influences, test emerging hypotheses in more diverse samples and inform the development of culturally responsive digital suicide prevention initiatives.

## Conclusions

Social media plays a complex role in the lives of young migrants, offering both protective and harmful influences on mental health. Most exposure to self-harm and suicide-related content appears to be accidental and can be distressing, while racist or discriminatory content may heighten feelings of isolation and increase suicide risk. At the same time, social media can support young migrants to form new connections, learn cultural norms and maintain ties with their country of origin, factors that can foster a sense of belonging and protect against suicide. It also provides an accessible, anonymous space to seek support, though the presence of poor-quality advice and stigma may deter some from seeking help. To maximise benefits and minimise harm, policymakers and healthcare providers should promote safe, culturally sensitive campaigns and content, and strengthen regulations and safety protocols to limit harmful content.

**Open peer review.** To view the open peer review materials for this article, please visit http://doi.org/10.1017/gmh.2026.10206.

**Supplementary materials.** The supplementary material for this article can be found at http://doi.org/10.1017/gmh.2026.10206.

**Data availability statement.** The data that support the findings of this study are available on request from the corresponding author [AB]. The data are not publicly available due to their containing information that could compromise the privacy of research participants.

**Author contribution.** AB was responsible for the interviews (including identifying and contacting participants, scheduling interviews and undertaking each interview), analysing the results and developing this manuscript. As an experienced qualitative researcher, ML joined the initial interviews to provide oversight and support to AB. SM helped to conceptualise the study, reviewed the manuscript multiple times, providing input to refine the codebook and themes. JR conceptualised the overarching study, obtained funding, designed the methodology, including the literature review and adaptation of the survey instrument, oversaw study sampling and data collection and analysis. GA helped to conceptualise the study and provided input into the manuscript through reviews.

**Financial support.** AB is a PhD student at the University of Melbourne whose research was supported by the Commonwealth through an Australian Government Research Training Program Scholarship [DOI: https://doi.org/10.82133/C42F-K220]. The overarching study was funded by the National Health and Medical Research Council (NHMRC) as part of an Investigator Grant awarded to JR (ID2008460). JR is also supported by a University of Melbourne Dame Kate Campbell Fellowship. GA is funded by an NHMRC Investigator grant (GNT2016501).

**Competing interests.** The authors declare none.

**Ethics statement.** The study received approval from the University of Melbourne Office of Research Ethics and Integrity (Reference number: Ref: 2023-27352-46587-3).

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
