## [Reviewer Report]

Thank you for the opportunity to review this important and timely manuscript exploring young migrants’ encounters with self-harm and suicide-related content on social media. The topic is highly relevant to Global Mental Health, particularly given the growing concerns around digital environments, migration-related vulnerabilities, and youth suicide prevention.

Overall, the paper is clearly written, methodologically sound, and offers valuable insights into an understudied area. The use of reflexive thematic analysis is appropriate, and the authors successfully integrate individual narratives with theoretical frameworks (Interpersonal Theory of Suicide, Cultural Theory of Suicide), strengthening the conceptual grounding of the findings. The manuscript makes a meaningful contribution to understanding how online spaces shape belonging, identity, and risk among young migrants, and it holds clear implications for policy and digital safety interventions.

Still, several areas could be strengthened to enhance clarity and impact. My detailed comments are below.

Major Comments

1. Conceptual framing of “young migrants” could be clarified: While the authors provide demographic data, the category “young migrants” remains broad. Given the heterogeneity in migration experiences, clarifying whether participants were international students, refugees, first-generation migrants, or other categories would deepen interpretation. Consider adding a short paragraph acknowledging variability in migration trajectories and how this may shape online experiences.

2. More depth is needed around the mechanisms of accidental exposure: Participants commonly report accidental exposure to self-harm content, but the manuscript could better articulate how this occurs (algorithmic pushes, shared content, platform-specific flows). A brief methodological reflection on platform mechanisms would strengthen policy implications.

3. Strengthen the link between themes and suicide prevention strategies: The Discussion section thoughtfully integrates theory but could further specify: how digital platforms might operationalise culturally responsive safety measures; which stakeholders (tech industry, clinicians, community organisations) should act on which findings. More concrete recommendations would increase the practical utility of the study.

4. Reflexivity is well described, but its influence on analysis could be elaborated: The authors discuss their “insider” migrant status, which is commendable.

To enhance transparency, consider adding one or two concrete examples of how reflexive discussions shaped theme development, coding decisions, or interpretation.

5. Sample limitations deserve slightly deeper treatment: The authors acknowledge several limitations, but one important point could be emphasised: “recruitment via a prior survey likely selected participants already willing to discuss self-harm and digital practices, potentially excluding migrants who avoid mental-health-related content due to stigma or cultural norms”. Clarifying this improves transferability assessment.

6. Opportunity to contextualise findings within the commercial determinants of child and adolescent mental health: The Discussion could be strengthened by briefly situating the study within the broader landscape of commercial influences shaping young people’s online environments. As noted by Carrasco JP, Estrella-Porter P, Cerame Á. Commodified upbringings (Int J Soc Psychiatry. 2025;71(6):1014–1029. doi:10.1177/00207640251341078), digital platforms operate through commercial logics that commodify attention and emotional engagement. Incorporating this perspective would help clarify how algorithmic systems, content monetisation, and platform economies may intersect with migration-related vulnerabilities to shape exposure to harmful content, offering a more robust structural interpretation of the findings.

Minor Comments

1. Table 2 is excellent but could benefit from concisely summarising potential risks and protective factors per theme; this would help readers capture theoretical implications at a glance.

2. Some quotations could be shortened or paraphrased to maintain narrative flow, particularly in Theme 1.

3. Consider briefly describing the age distribution of themes (if relevant), e.g., whether younger migrants (15–17) differed in their social media usage or emotional responses.

4. A brief statement on whether participants used predominantly specific platforms (e.g., TikTok vs. Instagram) may be useful for contextualising exposure patterns.

5. A few references (e.g., “in press”) should be updated prior to final publication if available.

---

## [Reviewer Report]

I appreciate the opportunity to review this manuscript and commend the authors for their research on this important topic. The study explores the young migrants' encounters with suicide and self-harm content, connection and seeking help on social media. Overall, I believe the study will significantly contribute to the literature and recommend its acceptance, albeit with minor revision. Below, I provide some comments and suggestions that I hope will strengthen the clarity and overall impact.

Introduction

1. If possible, I would recommend the authors to include data (prevalence) for the migrants engaging in self-harm/hospitalisation due to self-harm in Australia.

2. As the authors‘ are interested in how young migrants’ use social media for seeking help, I would recommend them including more research on how young people use social media to seek help, etc.

Results

1. While the authors are looking at social media as a whole, I wonder if it highlighting perhaps specific social media platforms where the young people are coming across this type of content would further add the nuance to the results.

2. There are themes here on belongingness and connection which should be discussed in the introduction section as well and how that perhaps links into seeking help.

3. It is unclear if young migrants were using social media to seek help from each other or other resources as well such as community pages, official websites such as Beondblue, etc. This needs to be flushed out more in this theme.

4. The authors briefly discuss that the young migrants were facing stigma when seeking help for self-harm. Was this happening in offline spaces or in online spaces or both? If they were experiencing this stigma then where were they turning to to seek help?

Overall, I would recommend the authors to consider rewriting some of the content of all themes. It appears to be very surface level and lacks in-depth reflection of the stories the participants are telling.

Discussion

1. There needs to be further in-depth discussion of young migrants' exposure to negative bits about their culture/country on social media impacted them and how that can perhaps lead to alienation and worsening mental health.

2. There needs to be some consideration given that experiencing stigma for seeking help is perhaps universal. Is there particular research which suggests that this phenomenon is more severe in migrant population? If yes, this needs to be further discussed. Further, I would have liked to see some more discussion about the anonymity afforded by social media in this aspect and how that may help young migrants. There is a bit of research in this area which can be used to add further to discussion.

3. The authors also need to consider how ban on social media for young people in Australia may impact everything. The ban is already in place and needs to be a part of the discussion here.

4. The strength and limitation section is very limited. The authors need to consider adding a bit more study nuance to this section.

---

## [Editor Report]

Dear Authors

Please see enclosed reviewer comments. We will be happy to accept this paper once you have addressed the comments from the two reviewers